# Unbiased Stochastic Proximal Solver for Graph Neural Networks with Equilibrium States

**Mingjie Li[1], Yifei Wang[2], Yisen Wang[1,3], Zhouchen Lin[1,3,4]** *
[1] National Key Lab of General Artificial Intelligence,
   School of Intelligence Science and Technology, Peking University
[2] School of Mathematical Sciences, Peking University
[3] Institute for Artificial Intelligence, Peking University
[4] Peng Cheng Laboratory

## Abstract

Graph Neural Networks (GNNs) are widely used deep learning models that can extract meaningful representations from graph datasets and achieve great success in many machine learning tasks. Among them, graph neural networks with iterative iterations like unfolded GNNs and implicit GNNs can effectively capture long-range dependencies in graphs and demonstrate superior performance on large graphs since they can mathematically ensure its convergence to some nontrivial solution after lots of aggregations. However, the aggregation time for such models costs a lot as they need to aggregate the full graph in each update. Such weakness limits the scalability of the implicit graph models. To tackle such limitations, we propose two unbiased stochastic proximal solvers inspired by the stochastic proximal gradient descent method and its variance reduction variant called USP and USP-VR solvers. From the point of stochastic optimization, we theoretically prove that our solvers are unbiased, which can converge to the same solution as the original solvers for unfolded GNNs and implicit GNNs. Furthermore, the computation complexities for unfolded GNNs and implicit GNNs with our proposed solvers are significantly less than their vanilla versions. Experiments on various large graph datasets show that our proposed solvers are more efficient and can achieve state-of-the-art performance.

## 1 Introduction

Graph Neural Networks (GNNs) (Zhou et al., 2020; Wu et al., 2020) can effectively aggregate information from its neighbors and then encode graph information into meaningful representations and have been widely used to extract meaningful representations of nodes in graph-structured data recently. Furthermore, Graph Convolution Networks (GCNs)(Kipf & Welling, 2016) involve the convolution structure in the GNNs and drastically improve the performance on a wide range of tasks like computer vision (Xu et al., 2020b), recommendation systems (He et al., 2020; Zhang et al., 2020b) and biochemical researches (Mincheva & Roussel, 2007; Wan et al., 2019). Due to these results, GCN models have attracted a lot of attention and various techniques have be proposed recently, including graph attention (Veličković et al., 2017), normalization (Zhao & Akoglu, 2019), linearization (Wu et al., 2019; Li et al., 2022) and others (Klicpera et al., 2018; Rong et al., 2020).

Current GNN models usually capture topological information of $T$-hops by performing $T$ iterations graph aggregation. However, $T$ cannot be large. Otherwise, their outputs may degenerate to some trivial points and such a phenomenon is called over-smoothing (Yang et al., 2020; Li et al., 2019). Therefore, traditional GNNs cannot discover the dependency with longer ranges. To tackle these problems, researchers have proposed some graph neural networks with iterative update algorithms (Yang et al., 2021a;b). The implicit graph neural networks (IGNNs) (Gu et al., 2020) is another type of such

---

*Corresponding author

model. Since these models will finally converge to an equilibrium state (stationary points or fixed points), we call these models **Graph Equilibrium Models** for convenience in the following paper.

Above graph equilibrium models enjoy superior advantages in capturing the long-range information because they implicitly finish the "huge hop" aggregation via its forward procedure. However, graph equilibrium models have to recursively aggregate the neighborhoods of graph nodes since solving their equilibrium state needs iteratively aggregating the full graph. Therefore, it needs expensive computation costs to deal with large graphs especially when they are dense. Although many works (Chen et al., 2018; Hamilton et al., 2017) propose different aggregation methods through the sampling nodes in traditional graph models, there are no guarantees for their convergence and unbiased approximation when applying them to the graph equilibrium models.

For the above reasons, how to efficiently obtain the outputs for these graph equilibrium models is an interesting problem worth exploring. Inspired by Yang et al. (2021b); Zhu et al. (2021); Zhang et al. (2020a)'s works which reveal the connections between the implicit and the unfolded graph neural networks' architecture and learnable graph denoising problems, we are trying to study the efficiency of the above models from the optimization view. Then we propose two stochastic solvers for these graph equilibrium models with convergence guarantees inspired by the stochastic proximal gradient descent algorithms. Since our forward procedure only needs to aggregate subgraphs, the proposed solvers are much more efficient than vanilla deterministic solvers by gradient descent or fixed-point iterations. Furthermore, we can theoretically prove that our solvers can obtain the unbiased output as the vanilla deterministic solvers do.

**Our Contributions.** We summarize the contributions of our methods as follows:

- By splitting the graph denoising optimization for the graph equilibrium models as several sub-optimization problems, we treat their forward procedure as solving the proper finite-sum optimization problem. Then we propose two stochastic solvers for graph equilibrium models: **Unbiased Stochastic Proximal Solver** (**USP solver**) and its variant with variance reduction **USP-VR solver**.

- Compared with the vanilla deterministic solvers which aggregate the full graph for graph equilibrium models' forward procedure, our USP solver and its variant only need to aggregate subgraphs to reach the equilibrium. Therefore, graph equilibrium models can be more efficient than before with our stochastic solvers.

- We theoretically prove that USP solvers can converge to the same outputs obtained by the vanilla deterministic forward procedure in expectation. Furthermore, we empirically demonstrate our proposed method's advantages with various experiments.

## 2 RELATED WORKS

### 2.1 GRAPH NEURAL NETWORKS

Most GNNs (Kipf & Welling, 2016; Veličković et al., 2017; Xu et al., 2018; Li et al., 2022) aggregate the graph information for finite times due to the over-smoothing problem. Thereby, they can hardly capture very long-range dependency. Contrary to these models, implicit graph models (Liu et al., 2021a; Gu et al., 2020; Park et al., 2021) aggregate the graph information for a lot of iterations or "infinite" iterations with theoretically non-trivial equilibrium outputs.

Moreover, recent works tries to explore the connections between the graph neural models and the graph denoising optimization problem. Some works (Zhu et al., 2021; Zhang et al., 2020a) recover different graph models to various graph denoising problems. Furthermore, Zhu et al. (2021) also proposed two types of models by reformulating graph denoising problems from spectral filters perspective. Some researchers focus on interpreting existing graph models from the view of redesigning the correlated graph denoising models (Yang et al., 2021a; Ma et al., 2021; Yang et al., 2021b).

Researchers also pay attention to GNNs from a different perspective. For example, robust graph neural networks (Jin et al., 2020; Luo et al., 2021), pretraining graph neural networks (Hu et al., 2019b; Qiu et al., 2020), explanations for graph networks (Ying et al., 2019; Yuan et al., 2020) and connections to differential systems (Xu et al., 2020a; Chamberlain et al., 2021; Wang et al., 2021).

## 2.2 Graph Equilibrium Models and implicit models

Since the implicit graph models (Gu et al., 2020; Chen et al., 2022) and unfolded graph Yang et al. (2021a); Liu et al. (2021c); Klicpera et al. (2018)'s forward procedure will converge to the equilibrium state of the corresponding optimization problem and Yang et al. (2021b) has already shown their relationships, we call these models the graph equilibrium models in the following. Yang et al. (2021a) briefly discusses the deterministic proximal gradient descent method's implementation for explicit graph models. Different regularizers' impacts on graph models' robustness have also been explored in Liu et al. (2021b;c) by adding them and reformulating their correlated optimization problems. Implicit graph models(El Ghaoui et al., 2021) are new types of graph neural architectures inspired by the deep equilibrium models (Bai et al., 2019; Li et al., 2021) in deep learning whose outputs are determined implicitly by the designation of fixed-point equations with graph aggregation. Since these models' forward procedures are all trying to reach the equilibrium points by iteratively aggregating on graphs, these models can capture the long-range information of graphs. Therefore, the graph equilibrium models can perform well on various large-scale graphs. However, as these models need aggregating whole graphs lots of times for the outputs, the computation complexity is large when dealing with large and dense graphs if we use the original solvers for graph equilibrium models.

## 2.3 Graph Neural Networks for Large Graphs

Aggregating through large or dense graphs needs huge computation complexity. Different training or inference strategy have been proposed to reduce the computation cost for training and testing procedures. GraphSAINT (Zeng et al., 2019) and Cluster-GCN (Chiang et al., 2019) propose subgraph sampling and restrict the back-propagation within the subgraphs to reduce the complexity of training. VR-GCN (Chen et al., 2017) uses the variance elimination method for training to improve the training performance. Narayanan et al. (2021) uses lazy update for weight parameters' training. GraphSAGE (Hamilton et al., 2017) updates a sub-sampled neighborhood of a node during training and inferencing to speed up GNNs.

## 3 Graph Equilibrium Models Using the Unbiased Stochastic Solver

### 3.1 Preliminaries and Settings

Implicit graph neural networks and other unfolded graph neural networks' forward procedure to get the output features after $n$ iterations $\mathbf{Z}^{(n)}$ for given input $\mathbf{X}$ can be formulated as follows:

$$\mathbf{Z}^{(n)} = \sigma\left(\mathbf{Z}^{(n-1)} - \gamma\mathbf{Z}^{(n-1)} + \gamma\mathbf{B} - \gamma\tilde{\mathbf{A}}\mathbf{Z}\mathbf{W}\mathbf{W}^\top\right), \tag{1}$$

with $\tilde{\mathbf{A}} = \mathbf{I} - \mathbf{D}^{-1/2}\mathbf{A}\mathbf{D}^{-1/2}$ denotes the Laplacian matrix, $\mathbf{A}$ is the adjacent matrix, input injection is $\mathbf{B} = \tilde{\mathbf{A}}\mathbf{X}\mathbf{W}_B$ as former works (Gu et al., 2020; Park et al., 2021) and $\sigma$ is the activation function. It can be viewed as the implicit graph neural networks when $\gamma = 1$ and otherwise it can be regarded as other unfolded graph neural networks. The above equation can also be regarded as solving the following graph denoising problems by deterministic gradient descent with step size $\gamma$,

$$\min_{\mathbf{Z}} G(\mathbf{Z}) = \min_{\mathbf{Z}} g(\mathbf{Z}) + f(\mathbf{Z}) = \min_{\mathbf{Z}}\left(\frac{1}{2}\|\mathbf{Z} - \mathbf{B}\|_F^2 + \frac{1}{2}tr\left((\mathbf{Z}\mathbf{W})^\top\tilde{\mathbf{A}}(\mathbf{Z}\mathbf{W})\right)\right) + f(\mathbf{Z}), \tag{2}$$

where $f$ is the proximal function that tries to constrain the output features and can induce the widely used activation functions by different choices. For example, if we choose to constrain the output features to be non-negative by setting $f$ as the positive indicator function, then the corresponding graph equilibrium models will use ReLU activation functions. Furthermore, the above formulation is the general formulation for different models. For example, IRLS (Yang et al., 2021a) sets $\mathbf{W} = \mathbf{I}$, APPNP (Klicpera et al., 2018) needs weight $\mathbf{W}$ and activation all be identity. Although our formulation Eqn (1) restricts the symmetric weight structure $\mathbf{W}\mathbf{W}^\top$, our optimization problem's weight parameter in Eqn (2) is not be constrained. Therefore, it won't have many impacts on the representative ability of the hidden problem and the graph equilibrium models, which are also illustrated in Hu et al. (2019a)'s work. To solve the above problem to get the equilibrium state,

most works apply the deterministic gradient descent method like Eqn (1). However, such a solver is time-consuming, especially when aggregating large graphs in each update. Therefore, we are going to propose new solvers for the above graph equilibrium models to make them more efficient. Before that, we are going to make some assumptions for the convenience of our analysis.

Firstly, we assume $\|\mathbf{WW}^\top\|_2 \leq 1/\|\tilde{\mathbf{A}}\|_2$ which can easily achieved by weight normalization (Salimans & Kingma, 2016) following Gu et al. (2020)'s setting. As shown in former works (Bai et al., 2020; Li et al., 2021), such a technique will not influence the performance much. Furthermore, we assume the derivative of $g(\mathbf{Z})$ are Lipschitz continuous with Lipschitz constant $L$ satisfying $1 < L < 2$, which can also be achieved by weight normalization. $G(\mathbf{Z})$ is strongly convex with $\mu = 1$, which can easily be achieved for many activation functions including ReLU. And in our work, we mainly focus on the graph equilibrium models without attention.

## 3.2 Solve Graph Equilibrium models via Unbiased Stochastic Solver

In order to obtain the solutions in a stochastic way, we first reformulate the graph optimization problem (2) from the view of edges as follows:

$$\min_{\mathbf{Z}} G(\mathbf{Z}) = \min_{\mathbf{Z}} g(\mathbf{Z}) + f(\mathbf{Z}) = \min_{\mathbf{Z}} \left( \frac{1}{2}\|\mathbf{Z} - \mathbf{B}\|_F^2 + \frac{1}{2} \sum_{(i,j) \in \mathcal{E}} \tilde{\mathbf{A}}_{ij}\|\mathbf{z}_i\mathbf{W} - \mathbf{z}_j\mathbf{W}\|_2^2 \right) + f(\mathbf{Z}),$$
(3)

where $\mathcal{E}$ is the full edges set of the graph, $(i,j) \in \mathcal{E}$ means an edge between node $i$ and $j$ exist in the edge set $\mathcal{E}$, $\mathbf{z}_i$ is the output feature vector for the $i$-th node and $\tilde{\mathbf{A}}_{ij}$ denotes $(i,j)$-th element of original Laplacian $\tilde{\mathbf{A}}$. Viewing graph edges as samples of the dataset, the original algorithm can be viewed as using the global proximal gradient descent method for the problem (3). However, acquiring the global gradient in each iteration is expensive as we demonstrate above. To make them more efficient, we convert the deterministic solvers for graph equilibrium models to the stochastic scheme.

First, we separate the global graph denoising objective $g$ into $m$ sub-objectives:

$$g_{\hat{\mathcal{E}}_k}(\mathbf{Z}) = \frac{1}{2}\|\mathbf{Z} - \mathbf{B}\|_F^2 + \frac{m}{2} \sum_{(i,j) \in \hat{\mathcal{E}}_k} \tilde{\mathbf{A}}_{ij}\|\mathbf{z}_i\mathbf{W} - \mathbf{z}_j\mathbf{W}\|_2^2,$$
(4)

where $\hat{\mathcal{E}}_k \subset \mathcal{E}$ is the $k$-th subset of edges which are randomly split into $m$ sets from the full graph set and each contains around $\lfloor \frac{|\mathcal{E}|}{m} \rfloor$ edges. Then the original optimization problem can be regarded as a finite sum optimization problem: $G(\mathbf{Z}) = \frac{1}{m}\sum_{k=1}^m g_{\hat{\mathcal{E}}_k}(\mathbf{Z}) + f(\mathbf{Z})$.

Then we can use the stochastic proximal gradient method by randomly choosing $k \in \{1, \cdots, m\}$ with the same probability and calculate the stochastic gradient for updating as follows:

$$\mathbf{V}^{(t)} = \mathbf{Z}^{(t)} + m\hat{\mathbf{A}}_{\hat{\mathcal{E}}_k}\mathbf{Z}^{(t)}\mathbf{WW}^\top - \mathbf{B},$$
(5)

$$\mathbf{Z}^{(t+1)} = \sigma\left(\mathbf{Z}^{(t)} - \eta_t\mathbf{V}^{(t)}\right),$$
(6)

where $\eta_t = \frac{1}{t}$ is the step size and $\hat{\mathbf{A}}_{\hat{\mathcal{E}}_k}$ is the laplacian matrix for $g_{\hat{\mathcal{E}}_k}$ with $m\mathbb{E}_k\hat{\mathbf{A}}_{\hat{\mathcal{E}}_k} = \tilde{\mathbf{A}}$ and its $(k_1, k_2)$-th element defined as:

$$\hat{\mathbf{A}}_{\hat{\mathcal{E}}_k}(k_1, k_2) = \begin{cases} \sum_{(k_1,j) \in \hat{\mathcal{E}}_k} \tilde{\mathbf{A}}_{k_1 j}, & \text{if } k_1 = k_2, \\ \tilde{\mathbf{A}}_{k_1 k_2}, & \text{if } (k_1, k_2) \in \hat{\mathcal{E}}_k, \\ 0, & \text{otherwise.} \end{cases}$$

We call the graph equilibrium models "A" using our solvers are Eqn (6) as the "A w. USP" in the following. "A w. USP" can converge to the equilibrium state $\mathbf{Z}^*$ as the following proposition states:

**Proposition 1.** *With the assumptions in Section 3.1, the expectations of our uniformly sampled stochastic gradient defined in Eqn (5) is the same as the $g(\mathbf{Z})$'s gradient, i.e. $\mathbb{E}_{\hat{\mathcal{E}}}\nabla_{\mathbf{Z}}\left[g_{\hat{\mathcal{E}}_k}(\mathbf{Z})\right] = \nabla_{\mathbf{Z}}g(\mathbf{Z})$. Furthermore, we can conclude that with our USP solver Eqn (6) and $\eta_t = \frac{1}{t}$, the result will converge to the solution of the global objective Eqn (3) in expectation with the sub-linear convergent rate, i.e., (We use $\mathbf{Z}^*$ to denote the minimizer of Eqn (3).)*

$$\mathbb{E}\left\|\mathbf{Z}^{(t)} - \mathbf{Z}^*\right\|_F^2 = O\left(\frac{1}{t}\right).$$
(7)

Proofs can be found in the Appendix. The above proposition indicates that iteratively forwarding the stochastic proximal gradient descent step Eqn (6) can finally get the solution for the problem Eqn (3), which means that our proposed stochastic solver is unbiased. Since the inner iterations Eqn (5) and (6) only need subgraphs instead of the full graph, our stochastic proximal solver can be much more efficient than the original solvers for graph equilibrium models in practice. However, the USP solver's forward procedure can only provide sub-linear convergence. Thereby, the outputs of our proposed solver will be less accurate than the vanilla deterministic models when their propagation times are limited during the forward procedure in practice and may influence performance. To solve such a problem, we propose another stochastic solver with the linear convergent rate in the following.

### 3.3 SOLVE GRAPH EQUILIBRIUM MODELS VIA UNBIASED STOCHASTIC SOLVER WITH VARIANCE REDUCTION

Although using the stochastic proximal gradient defined in Eqn. (5) is unbiased, the large variance of the stochastic gradient still hinders our stochastic solver's convergence speed. A straightforward way to reduce the variance is utilizing the full gradient to correct the gradient every $m_2$ iteration. For example, if next $m_2$ iterations of graph equilibrium models' forwarding are initialized with $\mathbf{Z}^{(0)} = \tilde{\mathbf{Z}}$, then for the $k$-th iteration with $k \geq 1$, we can use the modified gradient $\hat{\mathbf{V}}^{(k)}$ to replace the original $\mathbf{V}^{(k)}$ in Eqn (5):

$$\hat{\mathbf{V}}^{(k)} = \nabla g_{\hat{\mathcal{E}}_{i_k}}(\mathbf{Z}^{(k-1)}) - \nabla g_{\hat{\mathcal{E}}_{i_k}}(\tilde{\mathbf{Z}}) + \nabla g(\tilde{\mathbf{Z}}), \tag{8}$$

with $i_k$ sampled randomly from 1 to $m$ with the same probability. Moreover, we can easily conclude that the modified direction $\hat{\mathbf{V}}_k$ is unbiased and have a smaller variance than $\mathbf{V}^{(k)}$ in Eqn (5):

$$\mathbb{E}\left[\hat{\mathbf{V}}^{(k)}\right] = \nabla g(\mathbf{Z}^{(k-1)}),$$

$$\mathbb{E}\left\|\hat{\mathbf{V}}^{(k)} - \nabla g(\mathbf{Z}^{(k-1)})\right\|^2 \leq \mathbb{E}\left\|\mathbf{V}^{(k)} - \nabla g(\mathbf{Z}^{(k-1)})\right\|^2.$$

Replacing the original stochastic gradient with this modified version in the USP solver, we obtain our Unbiased Stochastic Proximal Solver with Variance Reduction (USP-VR) which is inspired by the variance reduction algorithm proposed in Xiao & Zhang (2014). The procedure for our USP-VR solver is listed in Alogrithm 1.

---

**Algorithm 1** USP-VR solver's procedure for the equilibrium state.

---

**Input:** Input graph Laplacian $\tilde{\mathbf{A}}$, input injection $\mathbf{B}$, initial state $\tilde{\mathbf{z}}_0$, step size $\eta$, maximum iteration number $m_1$, $i = 1$, maximum inner iteration number $m_2$, $\mathbf{W}, \sigma$ depend on the choices of different graph equilibrium models.

**Output:** output $\tilde{\mathbf{Z}}$.

1: Randomly split the whole edge set $\mathcal{E}$ into $m$ subsets $\{\hat{\mathcal{E}}_1, \cdots, \hat{\mathcal{E}}_m\}$ and generate the subgraph's laplacian matrix. $\{\hat{\mathbf{A}}_{\hat{\mathcal{E}}_1}, \cdots, \hat{\mathbf{A}}_{\hat{\mathcal{E}}_m}\}$.

2: **while** $i \leq m_1$ and $\tilde{\mathbf{Z}}^{(i)}$ not satisfies stop condition **do**

3:     $\tilde{\mathbf{Z}} = \tilde{\mathbf{Z}}^{(i)}$.

4:     $\tilde{\mathbf{V}} = \tilde{\mathbf{Z}} + \tilde{\mathbf{A}}\tilde{\mathbf{Z}}\mathbf{W}\mathbf{W}^\top - \mathbf{B}$.

5:     $\mathbf{Z}^{(0)} = \tilde{\mathbf{Z}}$.

6:     **for** $k = 1$ to $m_2$ **do**

7:         Randomly pick $s_k \in \{1, \cdots, m\}$ with the same probability.

8:         $\hat{\mathbf{V}}^{(k)} = (\mathbf{Z}^{(k-1)} - \tilde{\mathbf{Z}}) + m\hat{\mathbf{A}}_{\hat{\mathcal{E}}_{s_k}}(\mathbf{Z}^{(k-1)} - \tilde{\mathbf{Z}})\mathbf{W}\mathbf{W}^\top + \tilde{\mathbf{V}}$.

9:         $\mathbf{Z}^{(k)} = \sigma\left(\mathbf{Z}^{(k-1)} - \eta\hat{\mathbf{V}}^{(k)}\mathbf{W}\right)$.

10:     **end for**

11:     $\tilde{\mathbf{Z}}^{(i+1)} = \frac{1}{m_2}\sum_{k=1}^{m_2}\mathbf{Z}^{(k)}$.

12:     $i = i + 1$

13: **end while**

---

Since the global optimization problem $G(\mathbf{Z})$ is strongly convex, the USP-VR solver's multi-stage procedure can progressively reduce the variance of the stochastic gradient $\mathbf{V}_k$ and both $\tilde{\mathbf{Z}}, \mathbf{Z}^{(k)}$ will finally converge to $\mathbf{Z}^*$ with proper step size $\eta$.

**Proposition 2.** *With the same settings and assumptions in Proposition 1, we can conclude that our proposed USP-VR solver can converge to the equilibrium state with step size $\eta < \frac{1}{8L_{\max}}$ and sufficiently large $m_2$ so that*

$$\rho = \frac{8L_{\max}\eta^2(m_2+1)+1}{(\eta - 8L_{\max}\eta^2)m_2} < 1.$$

*where $L_{\max} = \max_k L_{g_{\hat{\mathcal{E}}_k}}$ where $L_i$ are the Lipschitz constants for $g_{\hat{\mathcal{E}}_i}$'s derivatives. The output $\tilde{\mathbf{Z}}^{(i)}$ after $i$ iterations will converge to the optimal solution $\mathbf{Z}^*$ linearly in expectation as follows:*

$$\mathbb{E}\left\|\tilde{\mathbf{Z}}^{(i)} - \mathbf{Z}^*\right\|_F^2 \le \rho^i \left\|\tilde{\mathbf{Z}}^{(0)} - \mathbf{Z}^*\right\|_F^2.$$

As the above proposition shows, our USP-VR solver enjoys a linear convergent rate like the vanilla deterministic solvers. Therefore, our USP-VR solver can achieve the solution which are more near to optimal compared with the USP-VR solver with limited propagation times in the forward procedure. Moreover, our USP-VR solver is still much faster than the original deterministic solvers since we only use the full graph for every $m_2$ iteration.

### 3.4 BACKPROPAGATION WHEN USING OUR SOLVERS

Assuming all the graph equilibrium models have achieved the equilibrium $\mathbf{Z}^{\text{f}}$ during its iterative forward pass, we use the one-step gradient by feeding $\mathbf{Z}^{\text{f}}$ to one deterministic iteration update (Eqn (1)) inspired by former works (Geng et al., 2021; Guo, 2013; Fung et al., 2022). Using $h$ to represent the deterministic iteration update (Eqn (1)) for convenience, the output for down-stream tasks is $\mathbf{Z}^{\text{o}} = h(\mathbf{Z}^{\text{f}}, \mathbf{A}, \mathbf{X})$. Since we can regard the whole structure as only propagating once from a near-optimal initialization point, we can only backward the final iteration to get the gradients. Gradients for learnable parameters in our methods can be written as follows:

$$\frac{\partial L(\mathbf{Z}^{\text{o}})}{\partial \theta} = \frac{\partial L(\mathbf{Z}^{\text{o}})}{\partial \mathbf{Z}^{\text{o}}} \left(\mathbf{I} + \frac{\partial h(\mathbf{Z}^{\text{f}}, \mathbf{A}, \mathbf{X})}{\partial \mathbf{Z}^{\text{f}}}\right) \frac{\partial h(\mathbf{Z}^{\text{f}}, \mathbf{A}, \mathbf{X})}{\partial \theta} \tag{9}$$

where $\theta$ denotes the learnable parameters for the graph equilibrium models $h$. The one-step gradient can be regarded as the first-order Neumann approximation of the original gradients with a more efficient memory cost. We adopt this approximated gradient in all our experiments.

## 4 EXPERIMENTS

In this section, we demonstrate the efficiency of our proposed USP and USP-VR solver compared with the traditional implicit models on large graph datasets with better performance on both node classification and graph classification tasks. Specifically, we test graph equilibrium models using our solvers against their vanilla version and other models on 6 popular node classification datasets (Flickr, Reddit, Yelp, PPI (Zitnik & Leskovec, 2017), OGBN (Hu et al., 2020) (Arxiv and Products)) and 2 graph classification datasets. These datasets are challenging for not only graph equilibrium models but also vanilla graph neural networks due to their sizes. We conduct the experiments on PyTorch (Paszke et al., 2019) with Torch-Geometric (Fey & Lenssen, 2019) and DGL (Wang et al., 2019). All the experiments are finished on an RTX-3090 GPU. The hyper-parameter settings and other details for the experiments are listed in the Appendix.

### 4.1 NUMERICAL EVALUATION

We first conduct an experiment to explore whether our proposed method can converge to equilibrium as our analysis shows. As we cannot obtain the closed-form solution for Problem (2), we use the gradient of $g(\mathbf{Z})$ in Eqn (3) as the evaluation of the convergence. This is because $\nabla_{\mathbf{Z}} g(\mathbf{Z}) = 0$ also demonstrates $\mathbf{Z}$ satisfies the first-order condition of Problem (2) if $\mathbf{Z}$ is obtained by our methods.

In this part, we use Reddit to draw the convergence curve for one layer IGNN, IGNN w. USP, and IGNN w. USP-VR layer with 128 output feature size for each node. We set $m = 20$ for our proposed solvers and $m_2 = 10$ for IGNN w. USP-VR. Then we draw the convergence curve of the relative gradient concerning the equivalent subgraph aggregation times (Each IGNN's iteration through the full graph is equivalent to $m$ times subgraph aggregation) as Figure 1 shows.

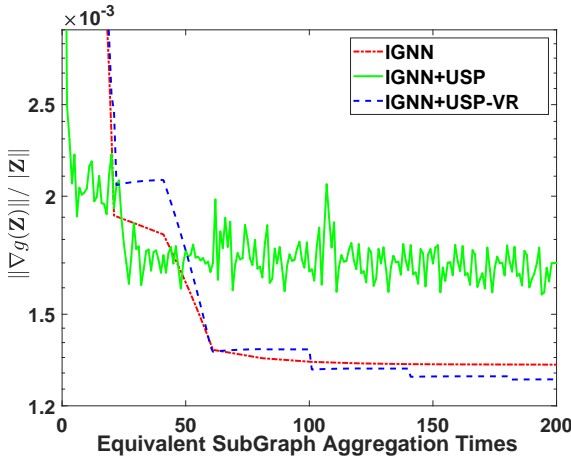

Figure 1: The convergence curve of each model's equilibrium state on Reddit Dataset.

The convergence curve is consistent with our analysis: IGNN w. USP can quickly converge compared with IGNN and IGNN w. USP-VR in the beginning while IGNN w. USP-VR will finally converge to a better solution compared with the other two models.

## 4.2 NODE CLASSIFICATION

**Comparison on popular large-scale graphs.** In this section, we first evaluate the performance of our methods on popular large-scale benchmark node classification graph datasets with the benchmark model APPNP and IGNN. The statistics of the four datasets are listed in Table 1. We compare the performance of the above models with their vanilla and our USP solvers. Apart from that, we also report the vanilla GCN and its sampling variants as baselines. All models share the uniform hidden dimensions of 256, $m = 20$ for our stochastic solvers, and other training details can be found in the Appendix. Furthermore, all models are trained without subgraph training except the results for the ClusterGCN and GraphSAINT. The results are listed in Table 2.

Table 1: Data Statistic ("m" stands for multi-label classification, and "s" for single-label).

| Dataset | Nodes | Edges | Degree | Classes | Training/Validation/Test |
|---|---|---|---|---|---|
| Flickr | $89,250$ | $899,756$ | 10 | $7(s)$ | $0.50/0.25/0.25$ |
| Reddit | $232,965$ | $114,615,892$ | 50 | $41(s)$ | $0.66/0.10/0.24$ |
| Yelp | $716,847$ | $13,954,819$ | 10 | $100(m)$ | $0.75/0.10/0.15$ |
| PPI (Multi Graph) | $14,755$ | $225,270$ | 14 | $121(m)$ | $0.79/0.11/0.10$ |
| OGBN-Arxiv | $169,343$ | $1,166,243$ | 7 | $40(s)$ | $0.54/0.18/0.18$ |
| OGBN-Product | $2,449,029$ | $61,859,140$ | 25 | $47(s)$ | $0.90/0.02/0.08$ |

From the table, one can see that our methods can outperform other vanilla models with notable advantages on most datasets, which means that our USP solvers can also capture long-range information such as APPNP and IGNN with their vanilla models. Moreover, the USP-VR solver is slightly better than USP solver because USP-VR can obtain outputs that are nearer to the optimal.

In addition to evaluating the prediction performance, we also draw the testing and training time for different models in Figure 2 to demonstrate the efficiency of our proposed solvers. From the figures, one can see that our USP solvers can accelerate APPNP and IGNN for more than $2\times$. Moreover, our USP solvers are more than $10\times$ faster than the vanilla solver on Reddit because its scale is larger than the other three datasets. Our USP-VR solver is slightly slower with better results than our USP solver since they need additional full graph aggregation in the forward procedure.

**Comparison on OGBN dataset.** Apart from the above datasets, we also conduct experiments for different models with their vanilla solvers and our USP solver on OGBN datasets listed in Table 3. From the table, one can see that our USP solvers can still return comparable results on OGBN datasets

Table 2: Comparison of test set Test Accuracy/F1-micro score with different methods. "OOT" here denotes that the time cost is more than $10\times$ longer than using USP solvers.

| Model | Flickr | Reddit | Yelp | PPI |
|---|---|---|---|---|
| GCN (Kipf & Welling, 2016) | $49.2 \pm 0.3\%$ | $93.3 \pm 0.1\%$ | $37.8 \pm 0.1\%$ | $51.2 \pm 0.3\%$ |
| GraphSAGE (Hamilton et al., 2017) | $50.1 \pm 1.3\%$ | $95.3 \pm 0.1\%$ | $63.4 \pm 0.6\%$ | $63.4 \pm 0.4\%$ |
| FastGCN (Chen et al., 2018) | $50.1 \pm 1.3\%$ | $95.3 \pm 0.1\%$ | $63.4 \pm 0.6\%$ | $51.3 \pm 3.2\%$ |
| ASGCN (Huang et al., 2018) | $50.1 \pm 1.3\%$ | $95.3 \pm 0.1\%$ | $63.4 \pm 0.6\%$ | $68.7 \pm 1.2\%$ |
| ClusterGCN (Chiang et al., 2019) | $48.1 \pm 0.5\%$ | $95.4 \pm 0.1\%$ | $60.9 \pm 0.5\%$ | $87.3 \pm 0.4\%$ |
| GraphSAINT (Zeng et al., 2019) | $51.5 \pm 0.1\%$ | $96.7 \pm 0.1\%$ | $64.5 \pm 0.3\%$ | $98.0 \pm 0.2\%$ |
| APPNP (Klicpera et al., 2018) | $51.2 \pm 0.2\%$ | $95.9 \pm 0.2\%$ | $63.2 \pm 0.2\%$ | $98.1 \pm 0.2\%$ |
| APPNP w. USP | $52.4 \pm 0.1\%$ | $96.2 \pm 0.2\%$ | $63.5 \pm 0.3\%$ | $97.9 \pm 0.2\%$ |
| APPNP w. USP-VR | $52.3 \pm 0.2\%$ | $96.4 \pm 0.3\%$ | $63.7 \pm 0.1\%$ | $98.2 \pm 0.3\%$ |
| IGNN (Gu et al., 2020) | $53.0 \pm 0.2\%$ | OOT | $65.8 \pm 0.2\%$ | $97.8 \pm 0.1\%$ |
| IGNN w. USP | $54.1 \pm 0.2\%$ | $96.7 \pm 0.3\%$ | $\mathbf{66.2 \pm 0.2}\%$ | $98.3 \pm 0.2\%$ |
| IGNN w. USP-VR | $\mathbf{54.3 \pm 0.1}\%$ | $\mathbf{96.8 \pm 0.2}\%$ | $66.1 \pm 0.2\%$ | $\mathbf{98.5 \pm 0.2}\%$ |

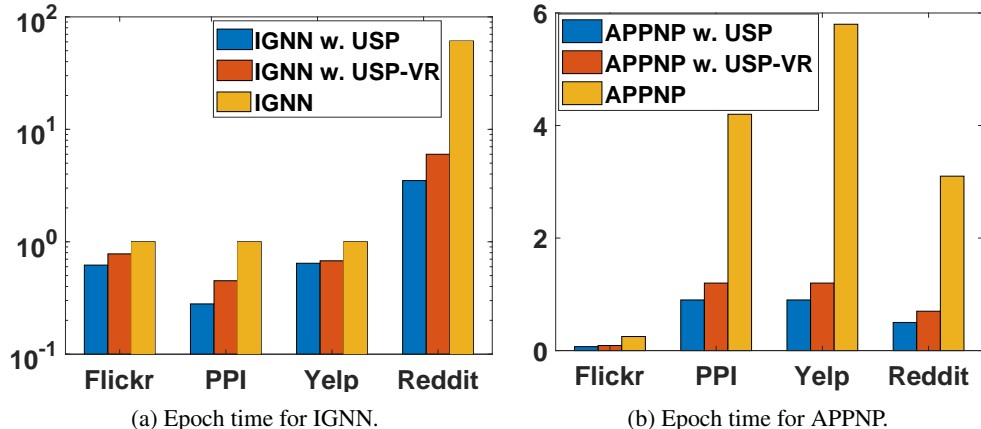

(a) Epoch time for IGNN.      (b) Epoch time for APPNP.

Figure 2: Comparison of the training time per epoch and test time through the full graph of IGNN and APPNP with and without our proposed solvers for different node classification datasets.

with much faster speed. From the above results, we can conclude that our proposed solvers can effectively improve the efficiency of various graph equilibrium models, especially on large graphs with comparable or even better performance on the node classification tasks.

| | | IGNN | IRLS | APPNP | DAGNN | JKNet |
|---|---|---|---|---|---|---|
| Arxiv | Accuracy | $70.4 \pm 0.8\%$ | $71.1 \pm 0.3\%$ | $71.1 \pm 0.2\%$ | $71.1 \pm 0.1\%$ | $71.0 \pm 0.2\%$ |
| | w. USP Accuracy | $\mathbf{72.7 \pm 0.2}\%$ | $\mathbf{71.6 \pm 0.2}\%$ | $\mathbf{71.4 \pm 0.2}\%$ | $\mathbf{71.3 \pm 0.1}\%$ | $\mathbf{71.5 \pm 0.4}\%$ |
| | w. USP Speedup | $\mathbf{3.5\times}$ | $\mathbf{2\times}$ | $\mathbf{2.5\times}$ | $\mathbf{2.3\times}$ | $\mathbf{2.4\times}$ |
| Product | Accuracy | $69.7 \pm 0.8\%$ | $73.8 \pm 0.2\%$ | $74.2 \pm 0.6\%$ | $73.7 \pm 0.6\%$ | $75.4 \pm 0.3\%$ |
| | w. USP Accuracy | $\mathbf{73.6 \pm 0.3}\%$ | $73.8 \pm 0.3\%$ | $\mathbf{74.6 \pm 0.5}\%$ | $73.5 \pm 0.7\%$ | $75.2 \pm 0.4\%$ |
| | w. USP Speedup | $\mathbf{5\times}$ | $\mathbf{3\times}$ | $\mathbf{6\times}$ | $\mathbf{6.2\times}$ | $\mathbf{4.5\times}$ |

Table 3: The empirical results for different Graph Equilibrium Models with their vanilla solver and our proposed USP solvers on OGBN datasets.

### 4.3 GRAPH CLASSIFICATION

Aside from the node classification, we also conduct experiments on graph classification datasets to see whether our methods can show consistent advantages on graph classification tasks. We use D&D (Dobson & Doig, 2003) (Bioinformatics Graphs) and COLLAB (Yanardag & Vishwanathan, 2015) (Social Graphs) for our experiments and use CGS and IGNNs as baselines. Training details can be found in the Appendix. Results are listed in Table 3a and test time for IGNN and our methods are drawn in Figure 3b. From the results, our proposed solvers also perform well.

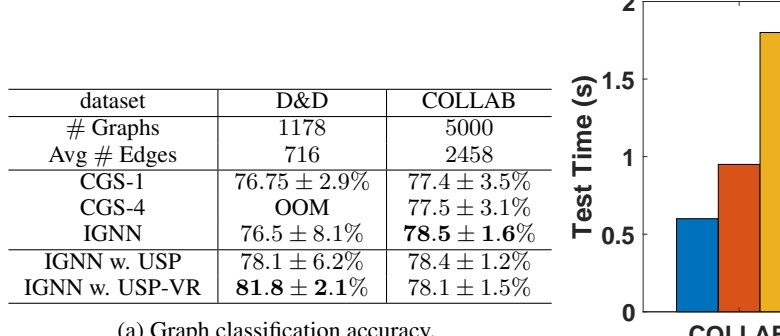

| dataset | D&D | COLLAB |
|---|---|---|
| # Graphs | 1178 | 5000 |
| Avg # Edges | 716 | 2458 |
| CGS-1 | $76.75 \pm 2.9\%$ | $77.4 \pm 3.5\%$ |
| CGS-4 | OOM | $77.5 \pm 3.1\%$ |
| IGNN | $76.5 \pm 8.1\%$ | $\mathbf{78.5 \pm 1.6}\%$ |
| IGNN w. USP | $78.1 \pm 6.2\%$ | $78.4 \pm 1.2\%$ |
| IGNN w. USP-VR | $\mathbf{81.8 \pm 2.1}\%$ | $78.1 \pm 1.5\%$ |

(a) Graph classification accuracy.

(b) Test time for IGNN and IGNN with our proposed solvers.

Figure 3: Graph classification accuracy and test time. Accuracies are averaged (and std are computed) on the outer 10 folds. CGS-$k$ denotes the CGS model with $k$ heads. "OOM" denotes out-of-memory.

IGNN with our proposed solvers is also faster than its origin in the experiment, $3\times$ on COLLAB and $1.5\times$ on D&D as the figure shows. We notice that our efficiency advantages are less significant than using our methods in the node classification tasks. The reason is that the scale of a single graph is much smaller in the graph classification task and the time cost of graph aggregations is not too large compared with others like linear layers' propagation or data processing. Furthermore, we can conclude that our stochastic solvers can perform more efficiently as the graph scale goes larger as our experiments show because the time cost of graph aggregation can gradually dominate the total cost of the implicit models as the graph gets larger.

### 4.4 Ablation Study: Comparison with former sampling methods.

First, we use the "SAGE" aggregator to directly replace APPNP's graph aggregation to conduct the experiments on OGBN-Arxiv. The results are shown in Table (4a). From the table, one can see that the original sage aggregator is not stable when applying them on APPNP because they cannot ensure convergence like our USP solvers.

| | Vanilla | with USP | with SAGE |
|---|---|---|---|
| Acc | $71.1 \pm 0.2\%$ | $\mathbf{71.4 \pm 0.2}\%$ | $67.8 \pm 5.7\%$ |
| Epoch Time | $4.1s$ | $\mathbf{1.7}s$ | $3.1s$ |

(a) APPNP variants with full-graph training.

| | GraphSAINT | GraphSAINT w. USP |
|---|---|---|
| Acc | $64.8 \pm 0.2$ | $\mathbf{65.2 \pm 0.3}\%$ |
| Epoch Time | $14s$ | $\mathbf{9.5}s$ |

(b) IGNN with sub-graph training on Yelp.

Figure 4: Comparison of different accelerating methods.

Furthermore, we conduct experiments using GraphSAINT on IGNN with and without our USP solvers in Table (4b) on Yelp. From the table, one can see that our USP solver can also accelerate the models when training them with the sub-graph techniques.

## 5 Conclusions

In our work, we first propose two stochastic solvers for the graph equilibrium models from the view of their relationship to the graph denoising optimization problem. Our proposed solvers can be applied to different graph equilibrium models such as APPNP and IGNN. Since our proposed solvers only need to propagate sub-graphs in each aggregation, our methods are much faster than the original fixed point iterative or gradient descent solvers used for implicit graph models and unfolded graph models, especially on large and dense graphs. Furthermore, we also theoretically prove that our solvers are unbiased and can finally output the same equilibrium state as the vanilla equilibrium models. The empirical results also demonstrate the advantages of different models with our proposed solvers for large-scale graphs.

ACKNOWLEDGMENTS

Zhouchen Lin and Yisen Wang were supported by National Key R&D Program of China (2022ZD0160302), the major key project of PCL, China (No. PCL2021A12), the NSF China (No. 62276004, 62006153), Open Research Projects of Zhejiang Lab (No. 2022RC0AB05), Huawei Technologies Inc., Qualcomm, and Project 2020BD006 supported by PKU-Baidu Fund.

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
