# OpenReview forum: "Unbiased Stochastic Proximal Solver for Graph Neural Networks with Equilibrium States"
_ICLR.cc/2023/Conference — ICLR 2023 poster_

### Official Review · Reviewer_1NkB · 2022-10-24

**Confidence:** 3
**Correctness:** 3
**Technical Novelty And Significance:** 3
**Empirical Novelty And Significance:** 3
**Recommendation:** 6

**Clarity, Quality, Novelty And Reproducibility:**

Clarity

It took me some time to understand the method on first reading because (1) looks like one layer of an ordinal GNN, although the main interest of this paper is implicit GNNs. Therefore, I would suggest briefly writing the formulation of implicit GNNs, and how (1) is derived.
Nevertheless, overall, the paper is well-written.


Quality

The proposed methods are justified from both theoretical and empirical points of view. First, Propositions 1 and 2 ensure that the proposed methods converge to an equilibrium state, demonstrating their theoretical soundness. Regarding the empirical evaluations, I want to clarify what the authors meant by computational complexity (e.g., in the abstract). Figure 2, Table 3, and Figure 3 certainly provide evidence for the improvement of numerical improvements in computational speed. They have also shown that there is no degradation in prediction performance. If the authors also intended space complexity, it would be desirable to evaluate memory usage quantitatively. That said, memory efficiency is implicitly evaluated, although I am not sure the authors intended this. For example, in Table 2, when using the Reddit dataset, the vanilla IGNNs caused OOM, while IGNN+USP and IGNN+USP-VR did not.

Novelty

If my understanding is correct, randomization is a relatively standard approach to reduce the computational complexity of optimization algorithms in expectation or with high probability. Also, variance reduction is a standard technique. In this sense, the method proposed in this paper is a natural improvement of the algorithm.

Reproducibility

The Appendix described the hyperparameters of the prediction model. Although this paper did not describe every full detail of dataset preparations, this is not a problem because references and dataset specifications are available. Since code is not provided, there is no guarantee of perfect reproduction. However, I think we can implement the code to reproduce the experiments to some degree.

Minor Comments

- P3 (1): $Z^{(n)}$, $Z^{(n+1)}$, $Z$ are undefined.
- P4 $\|WW^\top\| \leq 1/\tilde{A}$: this formula is invalid because the left hand side is scaler-valued, while the right hand side is vector-valued.
- P4 (3): $g(Z) + f(Z)$ -> $\min_{Z} g(Z) + f(Z)$
- P6 last equation: $\sum_{(k1, j) \in \tilde{\mathcal{E}}_k}$ -> $\sum_{(k_11, j) \in \tilde{\mathcal{E}}_k}$

**Details Of Ethics Concerns:**

N.A.

**Strength And Weaknesses:**

Strengths
- The proposed methods have guarantees for convergence.
- Numerical experiments demonstrated the usefulness of the proposed methods on large graph datasets.

Weaknesses
- Novelty in methodology is somewhat limited because randomization and variance reduction are relatively standard approaches to improve optimization algorithms.

**Summary Of The Paper:**

This paper proposed an efficient way to compute Graph Equilibrium models. The computation of the Graph Equilibrium model is an optimization problem of a graph denoising problem, and its naive solution is computationally expensive due to the use of an entire graph. This paper introduced the Unbiased Stochastic Proximal Solver (USP-solver) and its variance-reduction version USP-VR. They compute the Graph Equilibrium model by sampling the edges of the underlying graph. It was shown that USP converges to the equilibrium state at a sub-linear rate and USP-VR at a linear rate. Numerical experiments were conducted on node prediction and graph prediction problems to verify the practical usefulness of the proposed methods.

**Summary Of The Review:**

Although randomization and variance reduction are relatively standard approaches, as far as I know, these approaches have no application to implicit GNNs. In addition, the method is sound in that it has theoretical guarantees, and numerical experiments have shown its practical usefulness.

# Post-rebuttal comments

The responses by the authors solved my questions. Therefore, I want to keep my score, leaning to accept.

---

> ### Author Response · Authors · 2022-11-14
> **Response to Reviewer 1NkB**
>
> Thanks for your comments. Our responses are listed below.
>
> **1. About your concerns about our novelty.**
>
> We listed our contributions as follows:
> 1.  Our proposed solvers can effectively improve various graph equilibrium models' efficiency with comparable or even better results. To prove that, we added the experiments on OGBN dataset in our revised version as follows:
>
>     For OGBN-Arixv:
>
>     |    |  IGNN | IRLS | APPNP | DAGNN | JKNet
>     |  :----: | :----:  |  :----:  |  :----:  |  :----:  |  :----:  |
>     | Vanilla Accuracy | $70.4\pm0.8$%| $71.8\pm0.3$% | $71.1\pm0.2$% | $71.1\pm0.1$% | $71.0\pm0.2$% |
>     | w. USP Accuracy | $\mathbf{72.7\pm0.2}$% | $\mathbf{71.6\pm0.2}$% | $\mathbf{71.4\pm0.2}$% | $\mathbf{71.3\pm0.1}$% | $\mathbf{71.5\pm0.4}$% |
>     | w. USP SpeedUp | $\mathbf{3.5}\times$ | $\mathbf{2}\times$ | $\mathbf{2.5}\times$ | $\mathbf{2.3}\times$ | $\mathbf{2.4}\times$ |
>
>     For OGBN-Products:
>
>     |    |  IGNN | IRLS | APPNP | DAGNN | JKNet
>     |  :----: | :----:  |  :----:  |  :----:  |  :----:  |  :----:  |
>     | Vanilla Accuracy | $69.7\pm0.8$%| $73.8\pm0.2$% | $74.2\pm0.6$% | $73.7\pm0.6$% | $75.4\pm0.3$% |
>     | w. USP Accuracy | $\mathbf{73.6\pm0.3}$% | $73.8\pm0.2$% | $\mathbf{74.6\pm0.5}$% | $73.5\pm0.7$% | $75.2\pm0.2$% |
>     | w. USP SpeedUp | $\mathbf{5}\times$ | $\mathbf{3}\times$  | $\mathbf{6}\times$ | $\mathbf{6.2}\times$ | $\mathbf{4.5}\times$ |
>
>     From the results, one can see that our proposed solver can effectively speedup various graph equilibrium models.
>
> 2. Compared with other speedup methods by sampling, our proposed method is **unbiased and has the convergence guarantee**. Therefore can be more stable on graph equilibrium models. For example, directly using SAGE aggregation on APPNP will be unstable.
>
>     |    |  APPNP | APPNP with SAGE | APPNP with USP |
>     |  :----: | :----:  |  :----:  | :----:  |
>     | Accuracy | $71.1\pm0.2$% |  $67.8\pm5.7$% | $71.4\pm0.2$% |
>     | Time Cost | 4.1s | 3.1s | 1.7s |
>
>     For APPNP with SAGE, we use the Sage aggregation they proposed to replace the APPNP's graph aggregation. From the table, one can see that the unbiased property is important and can ensure our model performs stable on graph equilibrium models. In contrast to vanilla graph models, GraphSAGE may also perform well since they do not need to converge to the outputs which are near to proper optimal solutions like the graph equilibrium models.
>
> 3. We first proposed the stochastic optimization view to study the scalability of a class of graph neural networks.
>
> **2.  About your concerns on the memory cost.**
>
> We need to point out that there is no "OOM" in table 2, we use "OOT" to represent that the vanilla solver is more than $10\times$ slower than ours. Regarding the space complexity, we note that our model will need a slightly larger memory cost since we need to save the sampled graphs during propagation. But as in our Reddit experiment, our proposed solvers only need 1GB larger memory than IGNN (15GB against 14GB) if IGNN and our models both utilize the one-step gradient method.

---

> > ### Comment · Reviewer_1NkB · 2022-11-17
> > **Response to authors' comments**
> >
> > I thank the authors for giving us the response to my comment.
> >
> > **1. About your concerns about our novelty.**
> >
> > In my understanding, novelty is methodologically new things or experiment settings that have never been done in existing studies. Given this, I thought Parts 1 and 2 are what we think of as significance rather than novelty, although it may be just a matter of usage of words (I agree that Parts 1 and 2 are significant, see also the Quality section).
> > Regarding part 3, I understand that interpreting implicit GNNs as an optimization problem (rather than solving a fixed-point equation) is a new viewpoint (hence a novelty).
> >
> > **2. About your concerns about the memory cost.**
> >
> > > We need to point out that there is no "OOM" in table 2, we use "OOT" to represent that the vanilla solver is more than 10x slower than ours.
> >
> > I apologize that I mistakenly read the table. For the space complexity, I understand that memory overhead is approximately +7 %, which is acceptable.

---

### Official Review · Reviewer_S8SB · 2022-10-24

**Confidence:** 5
**Correctness:** 4
**Technical Novelty And Significance:** 2
**Empirical Novelty And Significance:** 2
**Recommendation:** 5

**Clarity, Quality, Novelty And Reproducibility:**

The whole paper is good, both in terms of clarity and quality. But the novelty is trivial in my opinion.

**Strength And Weaknesses:**

Strength:
1. The authors propose a method to accelerate the IGNN.

Weakness:
1. This work is incremental it only works for some specific frameworks (here is IGNN). However, there are so many GNNs like ODE and PDE GNN. If this type of incremental work makes it into the ICLR, we should have many for different GNNs in future. But does it inspire any member in this community? Is this really necessary?

**Summary Of The Paper:**

This work introduces the sampling way in the GNNs from optimization perspective. And thus it alleviates the computation cost on graph diffusion in different iterations. Furthermore, Variance Reduction (VR) is introduced for improving the performance.

**Summary Of The Review:**

According to my comments above, I am personally inclined to reject this paper. Of course I am happy to see other different opinions before making a decision.

---

> ### Author Response · Authors · 2022-11-14
> **Response to Reviewer S8SB**
>
> Thanks for your comments, our response are listed below.
>
> **1.  About your concerns on "our proposed methods' can only apply on few graph neural networks~(like IGNN)".**
>
> First, our proposed methods can not only apply to IGNN, but also can apply to various graph equilibrium models. Such GNN models can be regarded as iteratively solving optimization problems. As shown in [1,2], many widely used graph models can also be regarded as these models like IRLS, APPNP, DAGNN, JKNet. We added the experiments for APPNP, DAGNN, JKNet with our models in the revised version and we also listed them as follows.
>
> For OGBN-Arixv:
>
> |    |  IGNN | IRLS | APPNP | DAGNN | JKNet
> |  :----: | :----:  |  :----:  |  :----:  |  :----:  |  :----:  |
> | Vanilla Accuracy | $70.4\pm0.8$%| $71.8\pm0.3$% | $71.1\pm0.2$% | $71.1\pm0.1$% | $71.0\pm0.2$% |
> | w. USP Accuracy | $\mathbf{72.7\pm0.2}$% | $\mathbf{71.6\pm0.2}$% | $\mathbf{71.4\pm0.2}$% | $\mathbf{71.3\pm0.1}$% | $\mathbf{71.5\pm0.4}$% |
> | w. USP SpeedUp | $\mathbf{3.5}\times$ | $\mathbf{2}\times$ | $\mathbf{2.5}\times$ | $\mathbf{2.3}\times$ | $\mathbf{2.4}\times$ |
>
> For OGBN-Products:
>
> |    |  IGNN | IRLS | APPNP | DAGNN | JKNet
> |  :----: | :----:  |  :----:  |  :----:  |  :----:  |  :----:  |
> | Vanilla Accuracy | $69.7\pm0.8$%| $73.8\pm0.2$% | $74.2\pm0.6$% | $73.7\pm0.6$% | $75.4\pm0.3$% |
> | w. USP Accuracy | $\mathbf{73.6\pm0.3}$% | $73.8\pm0.2$% | $\mathbf{74.6\pm0.5}$% | $73.5\pm0.7$% | $75.2\pm0.2$% |
> | w. USP SpeedUp | $\mathbf{5}\times$ | $\mathbf{3}\times$  | $\mathbf{6}\times$ | $\mathbf{6.2}\times$ | $\mathbf{4.5}\times$ |
>
> From the results, one can see that our proposed solvers can accelerate various graph equilibrium models with notable speedup and comparable performances. Our proposed method can improve the efficiency of a class of models instead of only for IGNN.

---

### Official Review · Reviewer_XXk2 · 2022-10-25

**Confidence:** 4
**Correctness:** 4
**Technical Novelty And Significance:** 2
**Empirical Novelty And Significance:** 2
**Recommendation:** 6

**Clarity, Quality, Novelty And Reproducibility:**

The paper is clearly written with good quality. The idea is simple but effective.

**Strength And Weaknesses:**

# Strength:

The motivation and background of the studied problem have been clearly demonstrated, and the paper is well written. The paper revisits the implicit and optimization perspective of designing GNNs. The finite-sum reformulation draws a natural connection with stochastic proximal gradient algorithm and the variance-reduced variant. The idea is simple but effective and efficient as demonstrated in the experiments.

# Weakness

The forward computation of the solver is natural, simple, and well-justified. However, the backward computation is simply replaced by a one-step backward on the equilibrium point. This seems surprising if it can work well in practice. It would be great if the author can provide solid theoretical justification as well as the empirical study on the gradient approximation error due to this approximation. Moreover, the idea of sampling is standard in stochastic optimization. Therefore, the contribution and novelty are a bit weak.



**Summary Of The Paper:**

The paper revisits the implicit and optimization perspective of designing graph neural networks. Based on this perspective, the paper adopts two existing algorithms such as stochastic proximal gradient descent and its variance-reduced version to accelerate the forward computation with sampling. The backward computation is simply replaced by a one-step backward on the equilibrium point. Experiments demonstrate the effectiveness and efficiency of the proposed algorithm.


**Summary Of The Review:**

The paper provides a simple but effective approach for the training of large-scale implicit GNNs. The effectiveness and efficiency are clearly demonstrated. Solid theoretical and empirical justification of the backward computation can further improve the paper.

## After rebuttal

The revision improves the paper. I am happy to increase my score.

---

> ### Author Response · Authors · 2022-11-14
> **Response to Reviewer XXk2**
>
> Thanks for your comments. Our responses are listed below.
>
> **1.  About the one-step gradient.**
>
>
> We use the one-step gradient method in both IGNN and our model for the experiments, which only backward one forward iteration of IGNN with a full graph after reaching the equilibrium state as illustrated in [1]. The one-step gradient method can also be regarded as a first-order approximation to the exact gradient for the inverse of Jacobian using the Neumann series. Therefore, the error is high-order.
>
> If we use $\mathbf{Z}^{(n)} = f_\theta(\mathbf{Z}^{(n-1)},\mathbf{X},\mathbf{A})$ to represent the graph equilibrium models' $n$-th update iteration with $f_\theta$ here representing Eqn(1) and $\theta$ denoting the learnable parameters. Since the model will finally converge, we can use $\mathbf{Z} = f_\theta(\mathbf{Z};\mathbf{X},\mathbf{A})$ to constrain the output $\mathbf{Z}$. Then the gradient of $\theta$ with respect to loss L can be obtained by:
>
> $
> \begin{align}
>     \frac{\partial L}{\partial \theta} = \frac{\partial L}{\partial \mathbf{Z}} \frac{\partial \mathbf{Z}}{\partial \theta} = \frac{\partial L}{\partial \mathbf{Z}} (\frac{\partial f_\theta}{\partial \mathbf{Z}} \frac{\partial \mathbf{Z}}{\partial \theta} + \frac{\partial f_\theta}{\partial \theta})
> \end{align}
> $
>
> Then we can get,
>
> $
> \begin{equation}
>     \frac{\partial \mathbf{Z}}{\partial \theta} =  (\frac{\partial f_\theta}{\partial \mathbf{Z}} \frac{\partial \mathbf{Z}}{\partial \theta} + \frac{\partial f_\theta}{\partial \theta})
> \end{equation}
> $
>
> $
> \begin{equation}
>     (\mathbf{I}-\frac{\partial f_\theta}{\partial \mathbf{Z}})\frac{\partial \mathbf{Z}}{\partial \theta} = \frac{\partial f_\theta}{\partial \theta}
> \end{equation}
> $
>
> $
> \begin{equation}
>     \frac{\partial \mathbf{Z}}{\partial \theta}   = (\mathbf{I}-\frac{\partial f_\theta}{\partial \mathbf{Z}})^{-1}\frac{\partial f_\theta}{\partial \theta}
> \end{equation}
> $
>
> And the gradients can be rewritten as follows with the Neumann expansion
>
> $
> \begin{equation}
>     \frac{\partial L}{\partial \theta} = \frac{\partial L}{\partial \mathbf{Z}} (\mathbf{I}-\frac{\partial f_\theta}{\partial \mathbf{Z}})^{-1}\frac{\partial f_\theta}{\partial \theta} = \frac{\partial L}{\partial \mathbf{Z}} (\sum_{k=0}^\infty (\frac{\partial f_\theta}{\partial \mathbf{Z}})^{k})\frac{\partial f_\theta}{\partial \theta}
> \approx \frac{\partial L}{\partial \mathbf{Z}} (\mathbf{I} + \frac{\partial f_\theta}{\partial \mathbf{Z}})\frac{\partial f_\theta}{\partial \theta}\quad ({\rm{one}-\rm{step\ gradient}})
> \end{equation}
> $
>
> Furthermore,  the one-step gradient method (or the ``Jacobian Free'' method) can also obtain a useful gradient direction which can also make the model converge to the proper result as the theoretical analysis shown in [1].
>
>
> The experiments for IGNN on PPI with Phantom Gradient and implicit gradient also show that there is only a little difference between these two models' performance.
>
>
> |    |  IGNN | APPNP |
> |  :----: | :----:  |  :----:  |
> | Implicit Gradient | $97.6$%| $97.7$% |
> | One-step Gradient | $97.8$% | $98.1$% |
>
> However, the memory cost for the one-step gradient is quite small. Therefore, we use the one-step gradient in our paper as [1,2] does.
>
>
> [1] JFB: Jacobian-Free Backpropagation for Implicit Networks. Samy Wu Fung, Howard Heaton, Qiuwei Li, Daniel McKenzie, Stanley Osher, Wotao Yin
>
> [2] BPTT https://en.wikipedia.org/wiki/Backpropagation_through_time

---

> > ### Author Response · Authors · 2022-11-14
> > **Further Responses**
> >
> > **2.  About the contributions and novelty.**
> >
> > We listed our contributions as follows:
> > 1.  Our proposed solvers can effectively improve various graph equilibrium models' efficiency with comparable or even better results. To prove that, we added the experiments on the OGBN dataset in our revised version as follows:
> >
> >     For OGBN-Arixv:
> >
> >     |    |  IGNN | IRLS | APPNP | DAGNN | JKNet
> >     |  :----: | :----:  |  :----:  |  :----:  |  :----:  |  :----:  |
> >     | Vanilla Accuracy | $70.4\pm0.8$%| $71.8\pm0.3$% | $71.1\pm0.2$% | $71.1\pm0.1$% | $71.0\pm0.2$% |
> >     | w. USP Accuracy | $\mathbf{72.7\pm0.2}$% | $\mathbf{71.6\pm0.2}$% | $\mathbf{71.4\pm0.2}$% | $\mathbf{71.3\pm0.1}$% | $\mathbf{71.5\pm0.4}$% |
> >     | w. USP SpeedUp | $\mathbf{3.5}\times$ | $\mathbf{2}\times$ | $\mathbf{2.5}\times$ | $\mathbf{2.3}\times$ | $\mathbf{2.4}\times$ |
> >
> >     For OGBN-Products:
> >
> >     |    |  IGNN | IRLS | APPNP | DAGNN | JKNet
> >     |  :----: | :----:  |  :----:  |  :----:  |  :----:  |  :----:  |
> >     | Vanilla Accuracy | $69.7\pm0.8$%| $73.8\pm0.2$% | $74.2\pm0.6$% | $73.7\pm0.6$% | $75.4\pm0.3$% |
> >     | w. USP Accuracy | $\mathbf{73.6\pm0.3}$% | $73.8\pm0.2$% | $\mathbf{74.6\pm0.5}$% | $73.5\pm0.7$% | $75.2\pm0.2$% |
> >     | w. USP SpeedUp | $\mathbf{5}\times$ | $\mathbf{3}\times$  | $\mathbf{6}\times$ | $\mathbf{6.2}\times$ | $\mathbf{4.5}\times$ |
> >
> >     From the results, one can see that our proposed solver can effectively speedup various graph equilibrium models.
> >
> > 2. Compared with other speedup methods by sampling, our proposed method is **unbiased and has the convergence guarantee**. Therefore can be more stable on graph equilibrium models. For example, directly using SAGE aggregation on APPNP will be unstable.
> >
> >     |    |  APPNP | APPNP with SAGE | APPNP with USP |
> >     |  :----: | :----:  |  :----:  | :----:  |
> >     | Accuracy | $71.1\pm0.2$% |  $67.8\pm5.7$% | $71.4\pm0.2$% |
> >     | Time Cost | 4.1s | 3.1s | 1.7s |
> >
> >     For APPNP with SAGE, we use the Sage aggregation they proposed to replace the APPNP's graph aggregation. From the table, one can see that the unbiased property is important and can ensure our model performs stable on graph equilibrium models. In contrast to vanilla graph models, GraphSAGE may also perform well since they do not need to converge to the outputs which are near to proper optimal solutions like the graph equilibrium models.
> >
> > 3. We first proposed the stochastic optimization view to study the scalability of a class of graph neural networks.

---

> > > ### Comment · Reviewer_XXk2 · 2022-11-23
> > > **Further comments**
> > >
> > > Dear authors,
> > >
> > > Thank you for your detailed response. The revision improves the paper.
> > >
> > > One more suggestion is to provide a comprehensive summary of the GNN architectures that can be covered by the graph equilibrium model and provide detailed formulations of their graph signal denoising optimization perspective (for instance, it is unclear how JKNet, IRLS, and DAGNN are formulated as graph equilibrium model). This can help clarify when and how the proposed idea can be used.
> > >
> > > Overall, I am happy to increase my score.
> > >
> > > Reviewer XXk2

---

### Official Review · Reviewer_ejih · 2022-11-03

**Confidence:** 2
**Correctness:** 4
**Technical Novelty And Significance:** 2
**Empirical Novelty And Significance:** 3
**Recommendation:** 6

**Clarity, Quality, Novelty And Reproducibility:**

The paper does not make a very polished impression and is for that reason hard to follow from time to time.
It should be discussed more concretely why previous cannot be transferred to the setting of equilibrium because this does not seem to be apparent.
The proposed scheme seems relatively trivial but this can also be a feature.

**Details Of Ethics Concerns:**

I have no ethical concerns.

**Strength And Weaknesses:**

Strengths:
+ A simple, computationally less expensive to optimize graph equilibrium models is proposed.
+ The authors provide convergence guarantees of the average proposed sampling method and derive its convergence rate (O(1/t))
+ They also propose a simple variance reduction scheme that enjoys a linear convergent rate like the vanilla deterministic solvers.

Weaknesses and open questions:
- The claimed performance improvements in Table 2 are often not significant.
- How does the run time (in Figure 2) compare to explicit methods (which often do not perform much worse).
- The equilibrium solution $Z^*$ needs to be defined precisely. Currently, the formulation of Proposition 1 suggests that $Z^*$ might be unique, which is not the case in general. Also the experiments confirm this, since the IGNN variants do not seem to converge to the same solutions.
- How is the proposal different from (Chen et al., 2018; Hamilton et al., 2017) for "traditional graph models" and why should these not transfer to graph equilibrium models?
- The convergence results in expectation seem trivial as, in expectation, the procedure is not different from optimising the full model. The real challenge would lie in the analysis of the error that is introduced by the sampling scheme (or the average number of additional SGD steps to reach an equilibrium that are required for the actual sampling scheme and not its average).
- A complexity analysis of the proposed algorithm would strengthen the claim of computational advantages.

Points of minor critique and open questions:
- Examples for f on page 3 would be appreciated.
- Table 3 does not report any significance intervals.
- The formulation is restricted to symmetric weight structure $WW^T$ but the authors claim that this has no effect on the performance of the model. (It should impact its expressive power though...)
- In contrast to what the authors claim the weight normalisation will influence the training dynamics.
- page 5/6: `Therefore, our USP-VR solver can achieve a more accurate solution compared with the USP-VR solver with limited propagation times in the forward procedure.' does not seem to make sense.

**Summary Of The Paper:**

Two unbiased stochastic proximal solvers for learning graph equilibrium models are proposed. They are inspired by the stochastic proximal gradient descent method and its variance reduction variant (called USP and USP-VR solver). Both provide considerable computational speed-ups in comparison with the original solvers.


**Summary Of The Review:**

A strength of the paper is that the methodological proposals achieve performance speed-ups that are backed up by theoretical investigations.
It is not sufficiently clear to me, however, whether the contribution is sufficiently novel in comparison with similar schemes that have been developed for "non-equilibrium" models.
Furthermore, the presentation of the paper could be improved.

---

> ### Author Response · Authors · 2022-11-14
> **Response to Reviewer ejih**
>
> Thanks for your comments!
>
> **1.  The claimed performance improvements in Table 2 are often not significant.**
>
> The main contribution of our proposed method is that our model can accelerate the graph equilibrium model forward time as shown in Figure 2 and Table 3. From the results, one can see that our proposed solvers can accelerate various graph equilibrium models. As for the prediction performance, the graph equilibrium models with our solvers can also perform comparable performance on various graph datasets, especially IGNN with our solvers on Yelp and Flickr more than $2\%$ improvements compared with vanilla GCNs.
>
> **2.How does the run time (in Figure 2) compare to explicit methods (which often do not perform much worse).**
>
> Compared with IGNN, the GCN models are usually faster since they only use $2$ GCN layers and deeper GCN will become over-smoothing. However, IGNN can be regarded as a more than $5-10\times$ deeper GCN with shared weights. Therefore, IGNN performs better than GCN but can perform better since they can obtain long-range dependency but is around $5-10\times$ slower than GCN. But with our USP solvers, IGNN can show comparable speed as GCNs. The slow speed of IGNN is the motivation of our proposed solvers.
>
> Apart from IGNN, APPNP's speed is comparable with the vanilla GCNs and we can further accelerate them with our proposed solvers as our results are shown in Table 2 and Table 3.
>
> **3.  The equilibrium solution  $\mathbf{Z}^\*$  needs to be defined precisely. Currently, the formulation of Proposition 1 suggests that $\mathbf{Z}^\*$ might be unique, which is not the case in general. Also, the experiments confirm this, since the IGNN variants do not seem to converge to the same solutions.**
>
> $\mathbf{Z}^*$ means the exact solution for the optimization problem Eqn(2). As the optimization problem for graph denoising is convex, we can expect $\mathbf{Z}^*$ is unique. However, since obtaining the exact solution needs much more aggregations no matter which solver we use, there are differences in practical for $\mathbf{Z}^{(n)}$ ($\mathbf{Z}^{(n)}$ is the practical output after $n$ updates) and $\mathbf{Z}^*$. But since our solvers and the vanilla solvers will both converge to $\mathbf{Z}^*$ as $n$ increasing, the difference between our solvers and the vanilla solvers' outputs won't be large and will finally converge to $0$ as $n$ increasing.
>
> The slight experiment difference is also caused by the randomness of our graph sampling can improve the generalization abilities during training, like the dropout layers in most neural networks. In order to prove both solvers can converge to similar results, we also load the pre-trained APPNP on Flickr with our proposed method and then use the vanilla solver for testing as listed below:
>
> | Pretrained APPNP with USP   |  Accuracy |
> |  :----: | :----:  |
> | Test with vanilla solver | $52.1\pm0.3$% |
> | Test with USP sovler | $52.4\pm0.2$% |
>
> From the results above, one can see that using the vanilla solvers to evaluate the APPNP pretrained with our USP solvers can also return comparable results to the USP solvers. Moreover, the results are better than training and testing with the vanilla solvers ($51.2\%$). The results show that our USP solvers and the vanilla solvers can converge to similar outputs.
>
> **4.  How is the proposal different from (Chen et al., 2018; Hamilton et al., 2017) for "traditional graph models" and why should these not transfer to graph equilibrium models?**
>
> FastGCN (hen et al., 2018) samples nodes for each layer (each layer can be assumed as an optimization problem) and need more layers for the final prediction. But graph equilibrium models only optimize one optimization problem (one layer with iterative aggregations). Thereby, graph equilibrium models cannot use the FastGCN scheme.
>
> As for GraphSAGE (Hamilton et al., 2017), it uses node sampling and seems can be used in the graph equilibrium models. However, it has no convergence guarantee. Thus, as shown in Table 4(a) (also listed below), directly using the SAGE aggregation in APPNP won't return stable performance as the variance of ``APPNP with SAGE'' is much larger than APPNP with the vanilla solver and our USP solvers.
>
> |    |  APPNP | APPNP with SAGE | APPNP with USP |
> |  :----: | :----:  |  :----:  | :----:  |
> | Accuracy | $71.1\pm0.2$% |  $67.8\pm5.7$% | $71.4\pm0.2$% |
> | Time Cost | 4.1s | 3.1s | 1.7s |
>
> For APPNP with SAGE, we use the Sage aggregation they proposed to replace the APPNP's original graph aggregation. From the table, one can see that the unbiased property is important and can ensure our model performs stable on graph equilibrium models. Such a phenomenon is different from GraphSAGE's original paper since they are designed for the vanilla GCN models instead of Graph Equilibrium Models and the vanilla GCN does not need the final output to converge to certain states.

---

> > ### Author Response · Authors · 2022-11-14
> > **Further Responses**
> >
> > **5. About transferring the former sampling methods to graph equilibrium models.**
> >
> > For the sub-graph aggregating method, transferring them to graph equilibrium models won't return stable performance mainly because they lack the convergence guarantee as we illustrated in **Q4**.
> >
> > As for the sub-graph training method, it can save a lot of memory cost during training and also can be used in the graph equilibrium models even with our USP solvers as our experiments in Table 4(b) shows, we also listed below:
> >
> > |   Model | Accuracy | Training Time per Epoch |
> > |  :----: | :----:  | :----:|
> > | IGNN-SAINT | $64.8\pm0.2$%| $14$ s|
> > | IGNN-SAINT w. USP | $65.2\pm0.3$% | $9.5$ s|
> >
> > Our USP solver can further speed up the training procedure. Training with sub-graphs, the graph equilibrium models try to learn general modeling of graph denoising problems for sub-graphs' different graph denoising problem with trainable weights. Since we cannot obtain the whole graph in each training iteration, the performance is a little worse than directly using the whole graph.
> >
> > **6.  The convergence results in expectation seem trivial as, in expectation, the procedure is not different from optimizing the full model. The real challenge would lie in the analysis of the error that is introduced by the sampling scheme (or the average number of additional SGD steps to reach an equilibrium that are required for the actual sampling scheme and not it's average).**
> >
> > We are going to respond to your concern as follows:
> >
> > 1.  First, we want to note that the convergence guarantee is important for the graph equilibrium models. However, former sampling methods like GraphSAGE cannot ensure convergence and they cannot ensure the **unbiased** behavior. As our experiments are shown in **Q4**, GraphSAGE cannot stability converge and return unstable results on graph equilibrium models.
> >
> > 2.  As for the error rate, the difference between the output $\mathbf{Z}^{(n)}$ and $\mathbf{Z}^*$ is less than $O(1/n)$ for the USP solver and $O(\rho ^n)$ ($\rho$ here is the constant depended by the Lipschitz constant of the graph denoising problem) for the USP-VR (n here is the iteration number).
> >
> > 3. As for the sampling scheme: In our work, the sampling strategies may influence the stochastic model's convergence since it may influence the sub-problems Lipschitz constant. But the difference won't be large unless the sampled subgraph is too bad like the subgraph only contains limited nodes with each node having one neighbor. For example, the random walk sampler doesn't show any difference against uniform sampling in our work as listed in the table for SPGraphEq on Flickr (on RTX 3070). The reason why we choose a uniform sampler is that a random walk sampler needs more hyper-parameters to tune (batch size, step number, walk length).
> >
> > That is also a feature for our proposed solvers compared with former accelerating methods based on sampling.
> >
> > 4.  As for the practical convergence error for our model, as our experiments in Figure 1 shows, our proposed solver can quickly converge to the near-optimal solution. Therefore, our solver can return stable results on graph equilibrium models.
> >
> >
> > **7.  A complexity analysis of the proposed algorithm would strengthen the claim of computational advantages.**
> >
> > Our proposed solver only needs to aggregate $1/m$ subgraph in each aggregation. Therefore the computation cost is around $1/m$ for the graph aggregation. The difference between the theoretical and practical speedup is caused by the linear layers, activation layers, and other layers used in the graph equilibrium models.
> >
> >
> > **8.  Examples for f on page 3 would be appreciated.**
> >
> > We've added the example of $f$ in our revised version when using the ReLU non-linear activation function in the paper. When we use the ReLU activation function, we choose the positive indicator function, which returns $0$ if its input is positive and returns $\infty$ if the input is negative.
> >
> > If we use the leaky ReLU with parameter $\alpha$, then $f$ can be chosen as $\frac{1-\alpha}{2\alpha}x^2\mathbb{I}(x)$.

---

> > > ### Author Response · Authors · 2022-11-14
> > > **Further Responses**
> > >
> > > **9.  Table 3 does not report any significance intervals.**
> > >
> > > We've added them in the paper. Furthermore, we also add experiments for APPNP, DAGNN, JKNet (They are also graph equilibrium models due to the discussions in [1]) using our solvers in Table 3. The results are listed below:
> > >
> > > For OGBN-Arixv:
> > >
> > > |    |  IGNN | IRLS | APPNP | DAGNN | JKNet
> > > |  :----: | :----:  |  :----:  |  :----:  |  :----:  |  :----:  |
> > > | Vanilla Accuracy | $70.4\pm0.8$%| $71.8\pm0.3$% | $71.1\pm0.2$% | $71.1\pm0.1$% | $71.0\pm0.2$% |
> > > | w. USP Accuracy | $\mathbf{72.7\pm0.2}$% | $\mathbf{71.6\pm0.2}$% | $\mathbf{71.4\pm0.2}$% | $\mathbf{71.3\pm0.1}$% | $\mathbf{71.5\pm0.4}$% |
> > > | w. USP SpeedUp | $\mathbf{3.5}\times$ | $\mathbf{2}\times$ | $\mathbf{2.5}\times$ | $\mathbf{2.3}\times$ | $\mathbf{2.4}\times$ |
> > >
> > > For OGBN-Products:
> > >
> > > |    |  IGNN | IRLS | APPNP | DAGNN | JKNet
> > > |  :----: | :----:  |  :----:  |  :----:  |  :----:  |  :----:  |
> > > | Vanilla Accuracy | $69.7\pm0.8$%| $73.8\pm0.2$% | $74.2\pm0.6$% | $73.7\pm0.6$% | $75.4\pm0.3$% |
> > > | w. USP Accuracy | $\mathbf{73.6\pm0.3}$% | $73.8\pm0.2$% | $\mathbf{74.6\pm0.5}$% | $73.5\pm0.7$% | $75.2\pm0.2$% |
> > > | w. USP SpeedUp | $\mathbf{5}\times$ | $\mathbf{3}\times$  | $\mathbf{6}\times$ | $\mathbf{6.2}\times$ | $\mathbf{4.5}\times$ |
> > >
> > > From the results, one can see that our proposed solvers can accelerate various graph equilibrium models with notable speedup and comparable performances.
> > >
> > > **10.  The formulation is restricted to a symmetric weight structure but the authors claim that this has no effect on the performance of the model. (It should impact its expressive power though...)**
> > >
> > > It will not influence the final performance much since the representative power is enough for graph problems in our model. As we can see that in the graph denoising optimization problem:
> > >
> > > $
> > > \begin{equation}
> > > 	\min_\mathbf{Z} G(\mathbf{Z}) = \min_\mathbf{Z} g(\mathbf{Z}) + f(\mathbf{Z}) = \min_\mathbf{Z} \frac{1}{2}\|\mathbf{Z}-\mathbf{B}\|_F^2 + \frac{1}{2}tr\left(\left(\mathbf{Z}\mathbf{W}\right)^\top\mathbf{\tilde{A}}\left(\mathbf{Z}\mathbf{W}\right)\right) + f(\mathbf{Z}),
> > > \end{equation}
> > > $
> > >
> > > $\mathbf{W}$ here is unconstrained. Thereby, the optimization problem can regularize the output feature $\mathbf{Z}$ when projecting to various feature spaces by $\mathbf{ZW}$ with the graph Laplacian. The weight symmetric structure is shown when we calculate the gradient for the above problem. Therefore, the representative power is almost enough and won't influence the performance much. Moreover, as shown in [2], symmetric weight for other tasks with neural networks can also return comparable results.
> > >
> > > **11. About the weight normalisation will influence the training dynamics.**
> > >
> > > We apply the weight normalization to ensure the IGNN models are contractive. Otherwise, the IGNN won't converge as their paper [3] demonstrates. But as other graph equilibrium models usually assume $\mathbf{W}=\mathbf{I}$, the weight normalization is only used in IGNN.
> > >
> > > **11. page 5/6: `Therefore, our USP-VR solver can achieve a more accurate solution compared with the USP-VR solver with limited propagation times in the forward procedure.' does not seem to make sense.**
> > >
> > > ``accurate'' here means the solution is residuals for the output $\mathbf{Z}^{(n)}$ and the optimal solution for the graph denoising problem $\mathbf{Z}^*$ is smaller. As shown in Figure 1. We have updated such a sentence in the revised version.
> > >
> > > [1] Meiqi Zhu, Xiao Wang, Chuan Shi, Houye Ji, Peng Cui. Interpreting and Unifying Graph Neural Networks with An Optimization Framework.
> > >
> > > [2] Shell Xu Hu, Sergey Zagoruyko, and Nikos Komodakis. Exploring weight symmetry in deep neural network.
> > >
> > > [3] Fangda Gu, Heng Chang, Wenwu Zhu, Somayeh Sojoudi, Laurent El Ghaoui. Implicit Graph Neural Networks

---

> > > > ### Comment · Reviewer_ejih · 2022-12-06
> > > > **Acknowledgement of response**
> > > >
> > > > I thank the authors for their response and additional experiments. I have increased my score accordingly.
> > > >
> > > > However, I only lean towards an accept because the paper remains relatively unpolished and the contribution seems marginal for the following reasons:
> > > > (a) The proposed randomization is relatively standard to improve optimization algorithms and thus not based on a novel idea. (It is novel in the context of IGNNs though.)
> > > > (b) Direct methods are computationally more efficient. It is therefore questionable whether we should care about more efficient IGNNs. The main argument by the authors is that IGNNs can achieve higher accuracies than direct alternatives. However, note that the authors did not compare with the most performant variants that address issues like oversmoothing and can also handle deep GNNs. One could still argue that we might want to be able to optimize a diverse set of models and methods because of the no free lunch theorem. Because of that I am still willing to increase my score.

---

### Decision · Program_Chairs · 2023-01-20

**Decision:**

Accept: poster

**Justification For Why Not Higher Score:**

Novelty is somewhat incremental.

**Justification For Why Not Lower Score:**

Reviewers pointed out issues including relatively modest novelty, theoretical justification (e.g., on backward step) and limited applicability. The authors did a good job of response and pointed out that the model not only improves various graph equilibrium models' efficiency but also is unbiased and has the convergence guarantee. In addition, the authors provided suggested theoretical justification and showed broad applications with added experiments. Overall, the reviewers have more positive assessment after author responses.

**Metareview: Summary, Strengths And Weaknesses:**

This work proposed an efficient way to compute Graph Equilibrium models from optimization perspective, which for example reduce the computation cost on graph diffusion in different iterations. The computation of the Graph Equilibrium model is an optimization problem of a graph denoising problem, and its naive solution is computationally expensive due to the use of an entire graph. This paper introduced the Unbiased Stochastic Proximal Solver (USP-solver) and its variance-reduction version USP-VR. They compute the Graph Equilibrium model by sampling the edges of the underlying graph. Both provide considerable computational speed-ups in comparison with the original solvers. It was shown that USP converges to the equilibrium state at a sub-linear rate and USP-VR at a linear rate. Numerical experiments were conducted on node prediction and graph prediction problems to verify the practical usefulness of the proposed methods. Reviewers pointed out issues including relatively modest novelty, theoretical justification (e.g., on backward step) and limited applicability. The authors did a good job of response and pointed out that the model not only improves various graph equilibrium models' efficiency but also is unbiased and has the convergence guarantee. In addition, the authors provided suggested theoretical justification and showed broad applications with added experiments. Overall, the reviewers have more positive assessment after author responses.

**Note From Pc:**

if the above contains the word "oral" or "spotlight" please see: "oral" presentation means -> notable-top-5% and "spotlight" means -> notable-top-25%. As stated in our emails, we are disassociating presentation type from AC recommendations